# HAWAII: Hierarchical Visual Knowledge Transfer for Efficient Vision-Language Models

**Yimu Wang, Mozhgan Nasr Azadani, Sean Sedwards, Krzysztof Czarnecki**
University of Waterloo, Canada
{yimu.wang,mnasraza,sean.sedwards,k2czarne}@uwaterloo.ca

## Abstract

Improving the visual understanding ability of vision-language models (VLMs) is crucial for enhancing their performance across various tasks. While using multiple pretrained visual experts has shown great promise, it often incurs significant computational costs during training and inference. To address this challenge, we propose HAWAII, a novel framework that distills knowledge from multiple visual experts into a single vision encoder, enabling it to inherit the complementary strengths of several experts with minimal computational overhead. To mitigate conflicts among different teachers and switch between different teacher-specific knowledge, instead of using a fixed set of adapters for multiple teachers, we propose to use teacher-specific Low-Rank Adaptation (LoRA) adapters with a corresponding router. Each adapter is aligned with a specific teacher, avoiding noisy guidance during distillation. To enable efficient knowledge distillation, we propose fine-grained and coarse-grained distillation. At the fine-grained level, token importance scores are employed to emphasize the most informative tokens from each teacher adaptively. At the coarse-grained level, we summarize the knowledge from multiple teachers and transfer it to the student using a set of general-knowledge LoRA adapters with a router. Extensive experiments on various vision-language tasks demonstrate the superiority of HAWAII compared to popular open-source VLMs. The code is available at https://github.com/yimuwangcs/wise-hawaii.

## 1 Introduction

Vision-language models (VLMs) [1, 2] enable machines to perform complex reasoning over multimodal inputs by combining the powerful language reasoning capabilities of pretrained large language models (LLMs) [3, 4, 5] with the rich perceptual understanding offered by vision foundation models [6, 7, 8]. These two components are connected through alignment modules, such as Q-Formers [9] or MLP projections [10], which map visual tokens into a representation space compatible with LLMs. At the heart of this pipeline, the vision encoder plays a central role, as its ability to extract semantically rich visual features directly impacts the generation and reasoning capabilities of the VLM.

Recent studies have shown that incorporating multiple vision experts improves performance by a large margin [11, 12, 13, 14, 15]. Nevertheless, these gains in effectiveness often come at the cost of efficiency [16, 17, 18, 19, 20]: multi-expert setups require computing visual tokens from all vision experts during both training and inference, making them expensive and less practical for deployment, especially in latency-sensitive or resource-constrained settings [21, 22, 23]. As a result, there is growing interest in approaches that can retain the benefits of multiple vision experts while avoiding their substantial inference-time costs.

Knowledge distillation (KD) [24], as a general framework for transferring knowledge from a larger model (teacher) to a smaller model (student), has been widely used in various domains [25, 26, 27, 28]. As a pioneer study of KD in VLMs, MoVE-KD [29] distills knowledge from multiple visual experts

39th Conference on Neural Information Processing Systems (NeurIPS 2025).

into a single vision encoder using a *fixed set of Low-Rank Adaptation (LoRA) adapters [30]* for all teachers, enhancing visual understanding while only adding a small set of trainable parameters. However, learning from multiple teachers is challenging [31, 32], as the training data, model architecture, and training objectives of each teacher could be different. It can lead to noisy and redundant knowledge transfer, which can hinder the learning process with suboptimal performance [33].

To this end, we propose a novel **h**ierarchical visu**a**l kno**w**ledge tr**a**nsfer method for eff**i**c**i**ent VLMs, namely HAWAII. It is designed to distill knowledge from multiple visual experts, *i.e.*, SAM [6], ConvNext [34], EVA [8], and Pix2Struct [35], into a single vision encoder, specifically, CLIP's vision encoder, enabling it to inherit the complementary strengths of these experts with minimal computational overhead. HAWAII consists of a novel *mixture-of-LoRA-adapter* (MOLA) module and a *hierarchical knowledge distillation* (HKD) mechanism that enables the student encoder to distill knowledge at coarse-grained and fine-grained levels.

**Fine-grained distillation**. As each teacher's knowledge is different, due to the heterogeneity of training data, architecture, and optimization methods, in MOLA, teacher-specific LoRA adapters are employed to avoid conflicts between teachers' knowledge. Each adapter is aligned with its teacher separately, allowing the student encoder to learn from diverse teachers while mitigating noisy distillation. Moreover, to emphasize the informative tokens generated by each teacher, at the fine-grained level, HKD utilizes a new token importance scoring method, which assigns weights to tokens according to the similarity to the text instructions and visual features.

**Coarse-grained distillation**. To obtain the collective consensus among visual teachers, HKD summarizes the knowledge from multiple teachers using a projector. Then, MOLA incorporates a set of general-knowledge LoRA adapters and a router to align the student with the collective consensus for a global alignment.

In summary, the main contributions of this work are:

- We propose HAWAII, a novel framework that distills knowledge from multiple pretrained visual experts into a single vision encoder, improving the visual understanding ability of VLMs without incurring substantial computational overhead.

- The proposed MOLA module consists of teacher-specific LoRA adapters and general-knowledge LoRA adapters that enable the student encoder to learn from diverse teachers separately (fine-grained) and globally (coarse-grained), avoiding noisy and redundant knowledge transfer.

- HKD distills knowledge from multiple teachers at coarse-grained and fine-grained levels. At the fine-grained level, HKD utilizes teacher-specific LoRA adapters and token importance scoring to select and learn from the most informative tokens from each teacher, as indicated by the visual and text tokens. At the coarse-grained level, HKD summarizes the knowledge from multiple teachers and transfers it to the student encoder globally using general-knowledge LoRA adapters.

- Extensive experiments on various vision-language tasks [36, 37, 38, 39, 40, 41, 42, 43, 44, 45] show that HAWAII achieves better performance on all the benchmark datasets compared to the baseline model (LLaVA-1.5 [46]). In particular, the performance on VizWiz, SQA, and MMBench is improved by 7.8%, 5.5%, and 4.0%, respectively.

## 2 HAWAII

In this section, we introduce the HAWAII framework, which learns from multiple powerful visual teachers for a better visual perception ability. HAWAII inherits the complementary strengths of several experts without incurring substantial computational overhead. First, we introduce the architecture of HAWAII. Second, we present the details of MOLA, which consists of a set of teacher-specific and general-knowledge LoRA adapters in Section 2.2. Last, we provide the details of our hierarchical knowledge distillation method, which contains the coarse- and fine-grained distillation in Section 2.3.

### 2.1 Architecture

The overall architecture of HAWAII is presented in the upper part of Figure 1. It follows the general design (vision expert-projector-LLM) of existing MLLMs [2, 10, 47]. The vision expert is trained

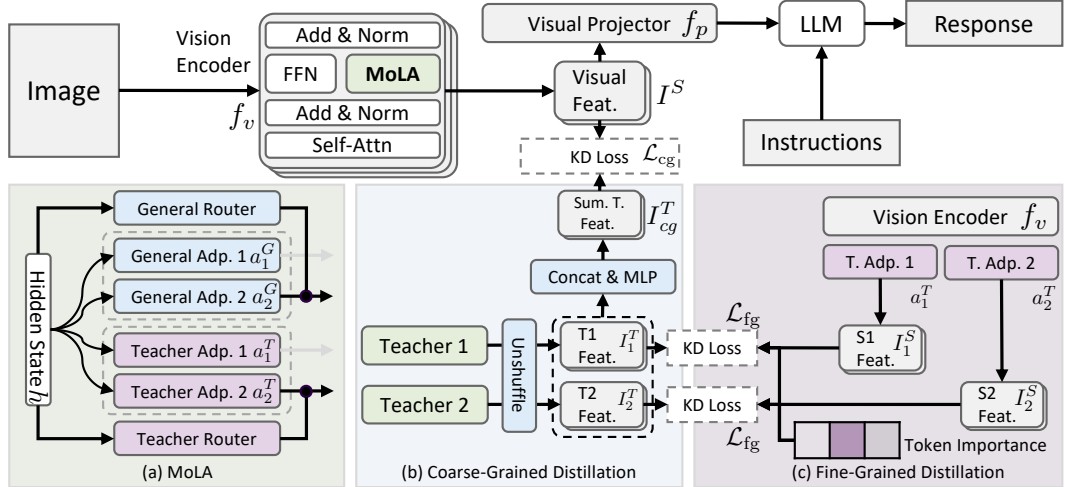

Figure 1: The overall architecture of HAWAII. We use two teachers for simplicity. (a) MoLA (Section 2.2) consists of teacher-specific LoRA adapters (Teacher Adp.) and general-knowledge LoRA adapters (General Adp.) with two routers controlling the activation of adapters. (b) Coarse-grained distillation (Section 2.3.1) first summarizes the knowledge from multiple teachers and then transfers it to the student encoder globally. "T1 Feat.", "T2 Feat.", and "Sum. T. Feat." represents the visual features $I_*^T$ generated by different teachers and the summarized teacher features $I_{cg}^T$. (c) In the fine-grained distillation (Section 2.3.2), teacher-specific LoRA adapters (T. Adp.) and token importance scoring (Figure 2) are employed to select and learn from the most informative tokens.

to distill knowledge from multiple pretrained vision experts and produces visual tokens used for visual comprehension. The projector maps the visual tokens to the LLM input space, and the LLM generates the instruction-following response.

The *vision encoder* $f_v(\cdot)$ takes the input image and generates a set of visual tokens $I^S \in \mathcal{R}^{m \times D}$, where $m$ is the number of visual tokens and $D$ is the dimension of each token. To boost the performance, instead of using multiple vision encoders [19, 47, 48], which would be computationally expensive, only one student vision encoder is employed. And, it is trained to distill knowledge from multiple pretrained vision experts [6, 7]. We introduce the mixture-of-LoRA-adapter (MoLA, Section 2.2) module that enables the student encoder to learn from diverse teachers in a fine-grained (Section 2.3.2) and coarse-grained (Section 2.3.1) manner.

A *visual projector* $f_p(\cdot)$ is applied to project the generated visual token $I^S$ to the LLM input space.

An *LLM* $f_{\text{LLM}}(\cdot)$ then takes the mapped visual tokens $f_p(I^S)$ and the textual instruction tokens $T$ as input to generate the instruction-following response $Y = \{y_i\}_{i \in [L]}$ as

$$p(Y|f_p(I^S), T) = \prod_{i=1}^{L} p(y_i|f_p(I^S), T, y_{<i}), \quad (1)$$

where $L$ is the length of the response and $y_{<i}$ is the previous tokens of $y_i$.

## 2.2 Mixture of LoRA Adapters

Directly fine-tuning the student encoder is challenging, as it often leads to overfitting on the limited fine-tuning data and catastrophic forgetting [29, 49]. To avoid this, we propose a mixture-of-LoRA-adapter (MoLA) module consisting of teacher-specific LoRA adapters and general-knowledge LoRA adapters [30] to enable the student encoder to learn from diverse teachers without forgetting. MoLA is illustrated in Figure 1 (a).

**Teacher-specific LoRA adapters.** Learning from multiple teachers is challenging [31, 32, 33], as each teacher [6, 7, 8] might have different training data, model architecture, and training objectives. Directly transferring diverse teachers' knowledge to the student could lead to noisy distillation and performance drop. To avoid this, we introduce a set of teacher-specific LoRA adapters $\{a_i^T\}_{i=1}^{N_t}$,

where $N_t$ is the number of teachers. Each adapter is designed to align with *one teacher* only, which avoids the conflicts between multiple teachers (see Section 2.3.2). Those adapters are applied to each feedforward layer of the student encoder $f_v(\cdot)$.

**General-knowledge LoRA adapters.** For learning the collective consensus from teachers and the training data, we introduce a set of general-knowledge LoRA adapters $\{a_i^G\}_{i=1}^{N_g}$ that are applied to each feedforward layer of the student encoder $f_v(\cdot)$, where $N_g$ is the number of general-knowledge LoRA adapters. The details of this general (global) knowledge transfer are provided in Section 2.3.1.

We adopt the general (sparse) design of mixture-of-experts (MoE) [50, 51] to select the LoRA adapters based on the hidden inputs of each layer. Specifically, we employ two sparse routers, *i.e.*, $f_r^T(\cdot)$ and $f_r^G(\cdot)$, to select the teacher-specific LoRA adapters and general-knowledge LoRA adapters, respectively. Formally, for each feedforward layer of the student encoder, the MoE output $F^*(\cdot)$ is computed as

$$F^*(h) = F(h) + a_i^T(h) + a_j^G(h),$$
$$\text{with } i = \text{ARGMAX}(f_r^T(h)) \text{ and } j = \text{ARGMAX}(f_r^G(h)), \tag{2}$$

where $h$ is the hidden input of the current layer and $F(\cdot)$ is the current layer. We denote the visual tokens generated by the student encoder with MOLA as $I^S$.

## 2.3 Hierarchical Knowledge Distillation

To integrate diverse teachers' knowledge into a single student encoder, we propose a hierarchical knowledge Distillation (HKD) mechanism that transfers knowledge at two levels of granularity, *i.e.*, coarse-grained and fine-grained levels. Specifically, for coarse-grained distillation (Section 2.3.1), we summarize the knowledge from multiple teachers (collective consensus) and transfer it to the student encoder globally. For fine-grained distillation (Section 2.3.2), teacher-specific LoRA adapters are employed to align with each teacher separately for a precise noise transfer. Moreover, to attend to the most informative tokens during knowledge transfer, we introduce a token importance scoring method (Figure 2) based on the similarity among teachers' visual tokens and the input instructions.

### 2.3.1 Coarse-Grained Distillation (CGKD)

To globally distill the knowledge from multiple teachers to the student encoder, we propose a coarse-grained distillation (CGKD) mechanism that first summarizes the knowledge from multiple teachers and then transfers it to the student encoder.

To obtain the collective consensus, *i.e.*, summarized teacher feature, each teacher's visual features are first unshuffled [2, 47, 52] to have the same length [2] as the student's visual features $I^S \in \mathcal{R}^{m \times D}$. Then, those visual tokens are channel-wise concatenated and the summarized feature $I_{cg}^T$ is obtained by applying a two-layer MLP $f_{cg}(\cdot)$ as

$$I_{cg}^T = f_{cg}\left(\text{CONCAT}\left(I_1^T, I_2^T, \ldots, I_{N_t}^T\right)\right) \in \mathcal{R}^{m \times D}, \tag{3}$$

where $I_i^T$ is the unshuffled visual tokens from the $i$-th teacher, and $\text{CONCAT}(\cdot)$ is the channel-wisely concatenation operation.

Next, we apply the coarse-grained distillation loss $\mathcal{L}_{cg}$ to transfer the collective consensus by minimizing the mean square error loss (MSE) between the summarized features $I_{cg}^T$ and the student encoder output $I^S$ as

$$\mathcal{L}_{cg} = \text{MSE}(I^S, I_{cg}^T). \tag{4}$$

### 2.3.2 Fine-Grained Distillation (FGKD)

Using LoRA adapters [30] to transfer knowledge from one teacher to a student has proven to be successful. However, transferring knowledge from multiple teachers to a single student is challenging, especially when using a fixed set of LoRA adapters [29] for all the teachers. The reason is that the noisy and redundant teachers' knowledge can hinder the learning process and lead to suboptimal performance [31, 32, 33], due to the conflicts among teachers, which arises from the heterogeneity of training data, architectures, and the training algorithms.

To address this challenge, we propose the fine-grained distillation (FGKD) that exploits teacher-specific LoRA adapters and token importance scoring. Each teacher-specific LoRA adapter is designed to align with one teacher only, allowing the student to learn from each teacher separately. Token importance scoring is used to select and attend to the most informative tokens from each teacher during knowledge transfer, reducing the noise and redundancy.

**Teacher-specific LoRA adapters.** We expect each teacher-specific LoRA adapter to learn the knowledge from one teacher *only*, such that the knowledge transfer is more effective and less noisy. We denote the output of the student encoder with only the $i$-th teacher-specific LoRA adapter $a_i^T$ being activated for each layer as $I_i^S$. Specifically, at each feedforward layer of the student encoder, we apply the LoRA adapter $a_i^T(\cdot)$ as $F(h) + a_i^T(h)$, where $F(\cdot)$ is the current layer and $h$ is the input to the layer. In that case, $I_i^S$ only needs to align with the $i$-th teacher's visual feature $I_i^T$, making the knowledge transfer procedure smooth and precise.

**Token importance scoring.** The key to knowledge distillation is to transfer the most important information [24]. As previous studies show that not all tokens are equally informative [18, 19, 20, 29], to identify the most informative tokens, we introduce a new similarity-based importance score that considers teachers' visual tokens and the input instructions $T$, allowing us to prioritize tokens that are more relevant to the task context. Specifically, for the $i$-th teacher, we compute the token importance score $s_i \in \mathcal{R}^{1 \times m}$ as

$$s_i = \text{MEAN}\left(\text{SOFTMAX}\left(\frac{\text{CONCAT}\left(\hat{I}_i^T, \hat{T}\right)(\hat{I}_i^T)^\top}{\sqrt{D}}\right)\right),$$
(5)

where $\hat{I}_i^T \in \mathcal{R}^{m \times D}$ and $\hat{T}$ are the visual tokens and input instructions projected by a learnable two-layer MLP to have the same dimension of the student features. $D$ is the dimension of the visual tokens.

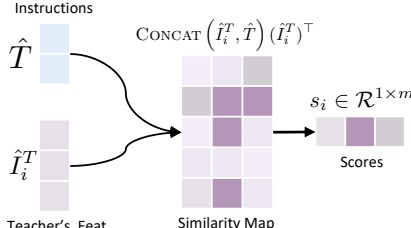

Figure 2: The calculation of token importance score $s_i$. To focus on the most informative tokens, we consider the similarity among the teacher's features and the input instructions $T$.

Now, with the token importance scores $\{s_i\}_{i \in [m]}$, the fine-grained distillation loss $\mathcal{L}_{\text{fg}}$ is calculated as

$$\mathcal{L}_{\text{fg}} = \frac{1}{N_t} \sum_{i=1}^{N_t} s_i \cdot \text{MSE}(I_i^S, \hat{I}_i^T).$$
(6)

## 2.4 Training Objectives

The overall training objective of HAWAII is to minimize the loss consisting of the text generation loss $\mathcal{L}_{\text{gen}}$ [4, 53], the coarse-grained distillation loss $\mathcal{L}_{\text{cg}}$ (Equation (4)), the fine-grained distillation loss $\mathcal{L}_{\text{fg}}$ (Equation (6)), and the MoE balance loss $\mathcal{L}_{\text{mb}}$ [51, 54, 55]. This is given as

$$\mathcal{L} = \mathcal{L}_{\text{gen}} + \lambda_1(\mathcal{L}_{\text{fg}} + \mathcal{L}_{\text{cg}}) + \lambda_2\mathcal{L}_{\text{mb}},$$
(7)

where $\lambda_1$ and $\lambda_2$ are the hyper-parameters to balance the losses. We set $\lambda_1 = 0.5$ and $\lambda_2 = 0.05$ for all our experiments.

# 3 Experiments

## 3.1 Experimental setup

**Implementation details.** We use Vicuna-v1.5-7B [3] as the LLM and use CLIP [1] for the vision encoder, with the teachers of CLIP (as CLIP is updated) being ConvNeXt [34], Pix2Struct [35], SAM [6], and EVA-02 [8]. The base version of HAWAII uses CLIP, ConvNeXt, and EVA-02 as the vision teachers, while for HAWAII $^\dagger$, we further add Pix2Struct as the teacher. To understand how different teachers contribute to the performance, we also conduct experiments with CLIP, ConvNeXt, EVA-02, and SAM as the teachers, denoted as HAWAII $^\ddagger$. The visual projector is a 2-layer MLP with the GELU activation function [56]. For MOLA, we use three (or four) teacher-specific LoRA adapters and three general-knowledge LoRA adapters for each FFN layer of the student encoder.

| Methods | VQA$^{\text{Text}}$ | VizWiz | GQA | SQA | POPE | MME | MMBench | MMMU | AI2D | SeedBench$^{\text{I}}$ |
|---|---|---|---|---|---|---|---|---|---|---|
| BLIP-2 [9] (Vicunna-13B) | 42.5 | - | - | 61.0 | 85.3 | 1293.8 | - | - | - | - |
| IDEFICS-9B [57] (LLaMA-7B) | 25.9 | 35.5 | 38.4 | - | - | - | 48.2 | - | - | - |
| Qwen-VL [58] (Qwen-7B) | 63.8 | 35.2 | 59.3 | 67.1 | - | - | 38.2 | - | - | 56.3 |
| mPLUG-Owl2 [59] (LLaMA-7B) | 54.3 | 54.5 | 56.1 | 68.7 | - | 1450.2 | 64.5 | - | - | 57.8 |
| Vicunna-1.5-7B | | | | | | | | | | |
| InstructBLIP [60] | 50.1 | 34.5 | 49.2 | 60.5 | - | - | 36.0 | 30.6 | - | 53.4 |
| Video-LLaVA [61] | 51.8 | 48.1 | 60.3 | 66.4 | 84.4 | - | 60.9 | - | - | - |
| MoVE-KD [29] | 58.3 | 52.3 | 63.2 | 69.4 | 86.9 | 1524.5 | 66.3 | - | - | - |
| LLaVA-1.5 [46] (Baseline) | 58.2 | 50.0 | 62.0 | 66.8 | 85.9 | 1510.7 | 64.3 | 34.7 | 55.5 | 66.1 |
| HAWAII | 58.7 | 53.9 | 62.8 | 70.5 | 87.3 | **1540.2** | 66.9 | 36.6 | 56.2 | 67.5 |
| Δ | ↑ 0.4 | ↑ 1.6 | ↓ 0.4 | ↑ 1.1 | ↑ 0.4 | ↑ 15.7 | ↑ 0.6 | ↑ 1.9 | ↑ 0.7 | ↑ 1.4 |
| HAWAII$^{\dagger}$ (HAWAII + Pix2Struct) | 58.6 | 54.2 | 63.6 | 69.5 | 86.8 | 1533.6 | 67.1 | 36.0 | 56.2 | 67.2 |
| HAWAII$^{\ddagger}$ (HAWAII + SAM) | **59.2** | **54.3** | 63.3 | 70.8 | 87.5 | 1528.6 | 66.7 | 36.9 | 55.8 | **67.9** |

Table 1: Performance comparisons of HAWAII and the baseline VLMs using Vicunna-1.5-7B (if not specified). HAWAII utilize CLIP, ConvNeXt, and EVA-02 as the teachers (the same setting with MoVE-KD), HAWAII$^{\dagger}$ further adds Pix2Struct as the teacher, and HAWAII$^{\ddagger}$ uses CLIP, ConvNeXt, EVA-02, and SAM as the teachers. The best results are in **bold** and the second best results are underlined.

Each adapter is a LoRA block [30] with rank of 32. The routers are sparse and 2-layer MLPs with the GELU activation function. Each router selects only the LoRA adapter with the highest probability. Models are run on eight NVIDIA A6000 GPUs with 48GB of memory.

**Training stages.** We follow the standard paradigm of LLaVA-1.5 [46]. The training of HAWAII consists of two stages, *i.e.*, pretraining and fine-tuning. The pretraining stage is to align the vision encoder with the LLM. During this stage, only the vision projector, LoRA adapters, and the routers are trained. The supervised fine-tuning stage is to align the vision encoder with the LLM and the instruction-following response. In this stage, the whole model is trained.

**Training datasets.** HAWAII uses the same training data as LLaVA-v1.5 [46]. Specifically, in the pretraining stage, we use 558K image-text pairs, while in the supervised fine-tuning stage, we use 665K instruction-following image-text data to boost the performance.

**Benchmarks and baselines**. We evaluate HAWAII on several image understanding tasks [36, 37, 38, 39, 40, 41, 42, 43, 44, 45]. Details are deferred to the Appendix. We compare HAWAII with several baseline methods, including general VLMs [9, 57, 58, 59, 60, 61] and a VLM with knowledge distillation [29].

## 3.2 Main Results

The results are shown in Table 1. Compared to the baseline method (LLaVA-1.5), HAWAII achieves significant improvements on most benchmarks, demonstrating its effectiveness. Results also demonstrate that compared to the existing knowledge distillation method [29] that uses the same teachers as HAWAII, HAWAII achieves better performance on most benchmarks, demonstrating the effectiveness of the proposed MoLA module and HKD mechanism.

## 3.3 Ablation Studies

In this part, we conduct ablation studies to analyze the effectiveness of the proposed components in HAWAII.

**Ablation on FGKD, CGKD, and MoLA.** The results are shown in Table 2. When all components are included, HAWAII achieves the best performance on most tasks (highlighted row), with an average of 63.7% across all tasks. The baseline model (LLaVA-1.5) with only FGKD (w/o token scoring) and teacher-specific LoRA adapters achieves 63.2% on average. Further adding the token importance scoring mechanism improves the performance to 63.5%. However, we also observe that the performance on GQA is slightly decreased, which might be due to the fact that GQA requires more general knowledge rather than specific knowledge from vision teachers. Adding CGKD and general-knowledge LoRA adapters further improves the performance to 63.7% on average.

**Number of visual teachers.** To understand how different teachers provide complementary knowledge for visual understanding, we conduct experiments with different teachers, as shown in Table 1. The

| Methods | VQA$^{\text{Text}}$ | VizWiz | GQA | SQA | POPE | MME | MMBench | MMMU | AI2D | SeedBench$^{\text{I}}$ | Avg. |
|---|---|---|---|---|---|---|---|---|---|---|---|
| LLaVA-1.5 | 58.2 | 50.0 | 62.0 | 66.8 | 85.9 | 1510.7 | 64.3 | 34.7 | 55.5 | 66.1 | 61.9 |
| + FGKD (w/ot token scoring) | 59.0 | 52.5 | **63.1** | 70.1 | 86.6 | 1532.1 | 66.8 | **36.7** | 54.6 | 66.3 | 63.2 |
| + token scoring | **59.1** | 52.5 | 62.8 | 70.2 | 87.4 | 1541.7 | **67.3** | 35.9 | 56.1 | 67.0 | 63.5 |
| + CGKD | 58.7 | **53.9** | 62.8 | 70.5 | 87.3 | 1540.2 | 66.9 | 36.6 | **56.2** | 67.5 | 63.7 |
| *w.* DoRA | 58.4 | 53.2 | 61.8 | 69.3 | **87.7** | **1558.5** | 66.9 | 35.2 | 55.5 | **67.8** | 63.4 |

Table 2: Ablation study on various vision-language tasks of HAWAII. We normalize the results of MME to compute the average results. FGKD and CGKD denote fine-grained distillation with teacher-specific LoRA adapters and coarse-grained distillation with general-knowledge LoRA adapters. W. DoRA represents the variant trained with DoRA for comparison.

| # | VQA$^{\text{Text}}$ | GQA | SQA | POPE | MME | MMMU | AI2D | SeedBench$^{\text{I}}$ |
|---|---|---|---|---|---|---|---|---|
| 1 | 58.7 | 62.6 | 70.1 | 84.5 | 1516.2 | 37.0 | 55.5 | 67.4 |
| 3 | 58.7 | 62.8 | 70.5 | 87.3 | 1540.2 | 36.6 | 56.2 | 67.5 |
| 5 | 58.6 | 62.8 | 70.4 | 85.2 | 1530.2 | 36.4 | 55.0 | 66.9 |

Table 3: Performance of HAWAII with different numbers of general-knowledge adapters.

| | VQA$^{\text{Text}}$ | GQA | SQA | POPE | MME | MMMU | AI2D | SeedBench$^{\text{I}}$ |
|---|---|---|---|---|---|---|---|---|
| LLaVA-1.5-13B | 61.3 | 63.3 | 71.6 | 85.9 | 1531.3 | 35.5 | 59.3 | 68.2 |
| MoVE-KD-13B | 59.7 | 64.2 | 73.2 | 85.7 | 1568.1 | - | - | - |
| HAWAII-13B | **61.7** | **64.7** | **75.0** | **86.6** | **1568.7** | **35.7** | **60.0** | **68.5** |

Table 4: Performance comparison using the Vicunna-1.5-13B.

| | VQA$^{\text{Text}}$ | GQA | SQA | POPE | MME | MMMU | AI2D |
|---|---|---|---|---|---|---|---|
| LLaVA-Next-7B | 64.9 | 64.2 | 70.1 | 86.5 | 1519.0 | 35.8 | 64.9 |
| MOVE-KD (LLaVA-Next-7B) | 63.7 | 64.5 | 70.7 | 86.7 | 1537.2 | - | - |
| HAWAII (LLaVA-Next-7B) | **65.5** | **65.2** | **72.0** | **87.8** | **1551.3** | **37.4** | **65.6** |

Table 5: Perofmance of HAWAII on LLaVA-Next-7B.

basic version of HAWAII uses CLIP, ConvNeXt, and EVA-02 as the teachers. Further adding Pix2Struct as the teacher improves the performance on VizWiz, GQA, and MMBench, compared to HAWAII. However, maybe due to the redundancy of knowledge, the performances on VQA$^{\text{Text}}$, SQA, and SeedBench$^{\text{I}}$ are slightly decreased. We further test the performance of HAWAII with CLIP, ConvNeXt, EVA-02, and SAM as the teachers, denoted as HAWAII [‡] in Table 1. Results show that HAWAII [‡] improves performance on VQA$^{\text{Text}}$, VizWiz, GQA, SQA, POPE, MMMU, and SeedBench$^{\text{I}}$, compared to HAWAII, as SAM might bring strong fine-grained descriptive visual understanding ability to the model. However, we also observe that the performance on MME decreases with adding more teachers, which might be due to the fact that MME requires more general common sense knowledge for reasoning rather than specific knowledge from vision teachers.

**Number of general-knowledge adapters.** The number of teacher-specific LoRA adapters is dependent on the number of visual teachers, whereas the number of general-knowledge LoRA adapters is a hyperparameter. To understand the optimal number of general-knowledge adapters, we present an ablation in Table 3. The results show that increasing the number of adapters to three improves performance on most benchmarks, while five adapters can lead to slight degradation, indicating that excessive redundancy may introduce overfitting.

**Generalizing to larger base models.** To test the efficiency of our proposed method, we conducted experiments with Vicunna-1.5-13B. The results in Table 4 show that HAWAII achieves significant improvements. Specifically, HAWAII improves the performance on SQA from 71.6 to 75.0. However, we also notice that with larger base models, the performance on POPE decreases as compared to that with a 7B model.

**The impact of the base method.** To understand how our proposed knowledge distillation generalizes across different base models, we conducted experiments with LLaVA-Next-7B [62]. The results in

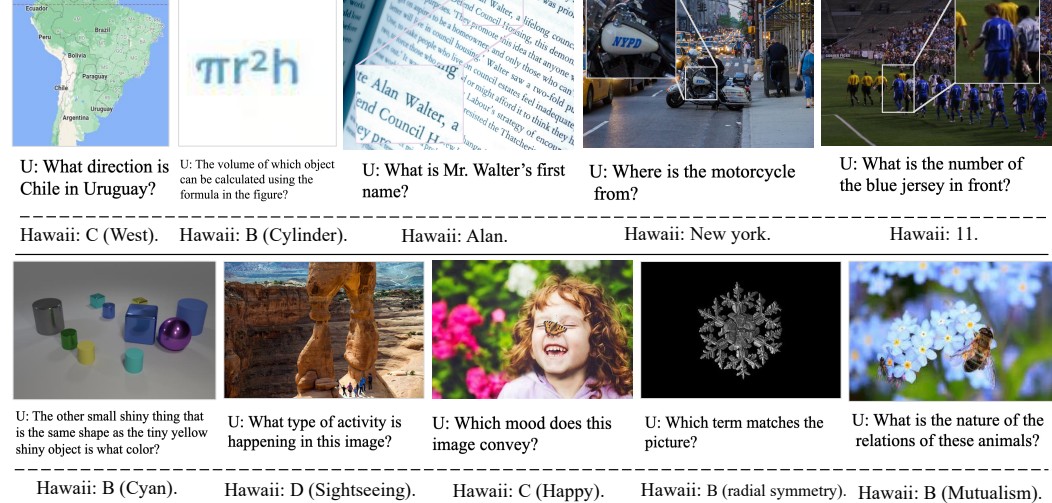

U: What direction is Chile in Uruguay?

U: The volume of which object can be calculated using the formula in the figure?

U: What is Mr. Walter's first name?

U: Where is the motorcycle from?

U: What is the number of the blue jersey in front?

Hawaii: C (West).  Hawaii: B (Cylinder).  Hawaii: Alan.  Hawaii: New york.  Hawaii: 11.

U: The other small shiny thing that is the same shape as the tiny yellow shiny object is what color?

U: What type of activity is happening in this image?

U: Which mood does this image convey?

U: Which term matches the picture?

U: What is the nature of the relations of these animals?

Hawaii: B (Cyan).  Hawaii: D (Sightseeing).  Hawaii: C (Happy).  Hawaii: B (radial symmetry).  Hawaii: B (Mutualism).

Figure 3: HAWAII is able to perform vision-language understanding tasks, such as emotion understanding, OCR, spatial reasoning, attribute reasoning, and relation reasoning. The examples are from the following benchmarks: VQA$^{\text{Text}}$ [37], MMBench [42], and SeedBench [45].

Table 5 show that HAWAII achieves significant improvements on most benchmarks, compared to the baselines.

**The impact of different LoRA methods.** We use LoRA in our design because of its generalizability. To understand how different LoRA adapters impact the performance, we conducted experiments with DoRA [63] replacing LoRA. The results are shown in Table 2. DoRA, which is more advanced than LoRA, is less generalizable than LoRA, as evidenced by the performance degradation on some benchmarks.

## 3.4 Qualitative Results

**Visualization of inference examples.** We perform qualitative evaluation to highlight the diverse reasoning capabilities of our model across a range of challenging visual understanding tasks [37, 42, 45]. As illustrated in Figure 3, HAWAII demonstrates strong attribute reasoning, accurately identifying fine-grained visual characteristics such as color, texture, and shape. For tasks involving OCR and mathematical content, the model effectively reads and interprets text in images. Beyond factual perception, HAWAII is capable of higher-level understanding, such as inferring image emotion and reasoning about contextual relationships and spatial arrangements. For instance, it can assess emotional tone from facial expressions and body language, and discern nature-related dependencies. These

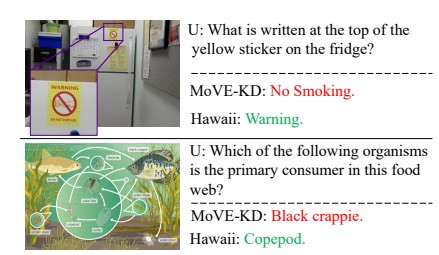

U: What is written at the top of the yellow sticker on the fridge?

MoVE-KD: No Smoking.

Hawaii: Warning.

U: Which of the following organisms is the primary consumer in this food web?

MoVE-KD: Black crappie.

Hawaii: Copepod.

Figure 4: Comparison between HAWAII and MoVE-KD [29] on OCR and visual-semantic reasoning capabilities.

examples showcase the model's comprehensive visual-language understanding, grounded in both low-level perception and abstract reasoning.

Moreover, a comparison with MoVE-KD [29] (Figure 4) highlights HAWAII 's stronger visual-semantic reasoning, as it accurately interprets ecological relationships in complex diagrams and effectively minimizes text hallucinations in OCR tasks.

**Visualization of token importance scores by different teachers and the instructions.** Different teachers and instructions typically attend to different regions of the image, providing diverse visual cues that are important for the model to develop a comprehensive understanding. To understand how the token importance scores distribute, we visualize similarity scores between different teachers and the instructions in Figure 5. As shown, the teachers and the instruction exhibit distinct preferences. Text instructions usually focus on the center objects in the images, which are usually indicated by the

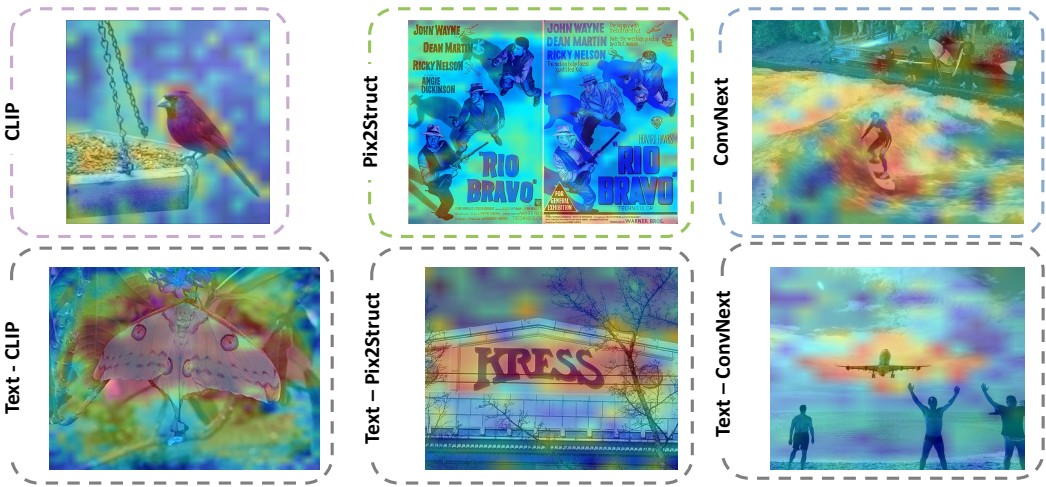

Figure 5: Visualization of the similarity score used in calculating importance score (Section 2.3.2) using HAWAII[†].

questions. For visual teachers, CLIP usually attends to the center objects, while ConvNext tends to care more about the common objects in an image (for example, people in the top right image). In contrast to CLIP and ConvNext, Pix2Struct focuses on the small signs and texts in the image, which is useful for OCR-related tasks.

## 4 Related work

**Multi-expert knowledge.** In the context of vision-language learning, multi-expert knowledge typically refers to the use of multiple pretrained visual models, each specialized in a particular domain or task, to provide richer and more diverse visual understanding. One common strategy for incorporating such knowledge is through auxiliary supervision or multitask learning [64, 65], where expert models trained on tasks such as segmentation, object detection, or depth estimation provide additional learning signals during training. These experts are typically integrated via auxiliary losses or parallel task-specific heads [66], allowing the model to benefit from complementary visual perspectives. While this approach has shown effectiveness, it often requires task-specific annotations and careful balancing of multiple objectives, which can complicate training and limit scalability.

Another strategy for incorporating multi-expert knowledge into vision components of VLMs involves using multiple visual encoders to extract diverse representations, which are then fused to form a unified visual understanding. These methods [13, 47, 48, 67] typically focus on efficiently integrating visual tokens generated by a mixture of pretrained visual experts. By drawing on the complementary strengths of these encoders, such approaches aim to enhance the model's visual perception capabilities. However, they often introduce substantial computational overhead due to the large number of visual tokens, particularly in approaches that concatenate token sequences [11, 13, 68]. In contrast, HAWAII adopts a different strategy by using multiple vision encoders as teachers to distill their knowledge into a single student encoder, enabling it to inherit their complementary strengths while maintaining efficiency.

**Knowledge distillation.** Knowledge distillation (KD) [24] is a process where a smaller, more efficient model called the student learns from the output logits or feature representations of a larger, pretrained model known as the teacher. In the context of vision-language learning, KD has been explored in several directions. Some approaches [69, 70] focus on distilling large vision-language models into smaller ones. This line of work aims to compress the knowledge of powerful multimodal models into more compact and efficient versions that can still perform effectively on vision-language tasks. In contrast to our work, these methods prioritize reducing the overall model size. Instead, our approach focuses on enhancing the visual capabilities of the vision encoder within a VLM by distilling knowledge from multiple expert teachers without necessarily reducing the VLM itself. Another common use of KD is to train efficient vision foundation models [7] by distilling smaller vision

backbones from larger teacher(s) in a standalone setting, separate from the VLM training pipeline. For example, InternViT-300M [71] is distilled from InternViT-6B using feature distillation with a cosine similarity loss applied between the hidden states of the final transformer layers. Similarly, RADIO [72] trains a vision model from scratch by merging multiple backbone models into a unified architecture through multi-teacher distillation. It employs feature-level distillation using cosine distance loss, with equal weighting applied to the outputs of each teacher. While effective, these approaches are highly computationally intensive and require massive datasets and substantial compute resources. In contrast to these standalone approaches, our work focuses on optimizing the student vision encoder within the training loop of a vision-language model, allowing it to benefit directly from multimodal supervision and alignment during training.

The work closest to ours is MoVE-KD [29], which distills knowledge from multiple visual experts into a single vision encoder using a weighted distillation loss with a fixed set of LoRA adapters [30]. The weights are shared between different teachers based on the attention weights from CLIP [1], which introduces a bias toward CLIP. In contrast, HAWAII introduce teacher-specific LoRA adapters which are aligned with each teacher separately, allowing the student encoder to learn from diverse teachers while avoiding noisy distillation. Moreover, the token importance scoring in HAWAII is based on each teacher's visual features and the input instructions, which helps to select the most informative tokens from each teacher without introducing bias toward any specific teacher.

## 5    Conclusion, Limitations, and Societal Impacts

**Conclusion.** We introduced HAWAII, a novel framework that distills knowledge from multiple pretrained visual experts into a single vision encoder. HAWAII consists of a novel mixture-of-LoRA-adapter (MOLA) module and a new hierarchical knowledge distillation (HKD) mechanism. MOLA consists of teacher-specific LoRA adapters and general-knowledge LoRA adapters that enable the student encoder to learn from diverse teachers while learning general knowledge from the training data. HKD distills knowledge from multiple teachers at coarse-grained and fine-grained levels. The coarse-grained distillation summarizes the knowledge from multiple teachers and transfers it to the student encoder globally. The fine-grained distillation utilizes teacher-specific LoRA adapters and token importance scoring to select the most informative tokens from each teacher for distillation. Extensive experiments on various vision-language tasks demonstrate the superiority of HAWAII over existing methods with minimal computational overhead.

**Limitations**. Due to the limitation of computational resources, we only used five pretrained vision experts in our experiments. Also, we only evaluated HAWAII using the Vicuna-v1.5-7B [3] as the LLM. In the future, it would be interesting to explore the performance of HAWAII with more pretrained vision experts and different LLMs. We only distill knowledge from the visual experts to the vision encoder, while the knowledge distillation from a bigger LLM to a smaller LLM is not considered. We believe that further improvements can be achieved by distilling knowledge from a bigger LLM to a smaller LLM.

**Societal impacts and safeguards.** The proposed HAWAII framework is designed to enhance the performance of VLMs. Thus, it inherits the same societal impacts as existing VLMs. The use of HAWAII and VLMs in general may raise concerns related to bias, misinformation, and privacy. However, we have taken steps to mitigate these risks by carefully curating the training data and implementing safeguards to ensure responsible use.

## Acknowledgement

This work was supported by the Natural Sciences and Engineering Research Council of Canada (NSERC)-CSE Research Community project entitled "An End-to-End Approach to Safe and Secure AI Systems" and NSERC's Postdoctoral Fellowship. Researchers funded through the NSERC-CSE Research Communities Grants do not represent the Communications Security Establishment Canada or the Government of Canada. Any research, opinions, or positions they produce as part of this initiative do not represent the official views of the Government of Canada.

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

# A Experiments

## A.1 Benchmark Datasets

We evaluate our method on the following benchmark datasets: MME [41], MMBench [42], Seed-Bench [45], GQA [38], SQA [39], MMMU [43], POPE [40], AI2D [44], VizWiz [36], and TextVQA [37].

**MME [41].** The MME benchmark is designed to rigorously evaluate a model's perceptual and cognitive abilities through 14 subtasks. It employs carefully constructed instruction-answer pairs and concise instructions to minimize data leakage and ensure fair evaluation. This setup provides a robust measure of a model's performance across various tasks.

**MMBench [42].** MMBench offers a hierarchical evaluation framework, categorizing model capabilities into three levels. The first level (L-1) focuses on perception and reasoning. The second level (L-2) expands this to six sub-abilities, while the third level (L-3) further refines these into 20 specific dimensions. This structured approach allows for a nuanced and comprehensive assessment of a model's multifaceted abilities.

**Seed-Bench [45].** SEED-Bench consists of 19K multiple-choice questions with accurate human annotations, covering 12 evaluation dimensions including both the spatial and temporal understanding.

**GQA [38].** GQA is structured around three core components: scene graphs, questions, and images. It includes not only the images themselves but also detailed spatial features and object-level attributes. The questions are crafted to assess a model's ability to comprehend visual scenes and perform reasoning tasks based on the image content.

**ScienceQA [39].** ScienceQA spans a wide array of domains, including natural, language, and social sciences. Questions are hierarchically categorized into 26 topics, 127 categories, and 379 skills, providing a diverse and comprehensive testbed for evaluating multimodal understanding, multi-step reasoning, and interpretability.

**MMMU [43].** MMMU includes 11.5K meticulously collected multimodal questions from college exams, quizzes, and textbooks, covering six core disciplines: Art & Design, Business, Science, Health & Medicine, Humanities & Social Science, and Tech & Engineering. These questions span 30 subjects and 183 subfields, comprising 30 highly heterogeneous image types, such as charts, diagrams, maps, tables, music sheets, and chemical structures.

**POPE [40].** POPE is tailored to assess object hallucination in models. It presents a series of binary questions about the presence of objects in images, using accuracy, recall, precision, and F1 score as metrics. This approach offers a precise evaluation of hallucination levels under different sampling strategies.

**AI2D [44].** AI2D is a dataset of over 5000 grade school science diagrams with over 150000 rich annotations, their ground truth syntactic parses, and more than 15000 corresponding multiple choice questions.

**VizWiz [36].** VizWiz consists of over 31,000 visual questions originating from blind people who each took a picture using a mobile phone and recorded a spoken question about it, together with 10 crowdsourced answers per visual question.

**TextVQA [37].** TextVQA emphasizes the integration of textual information within images. It evaluates a model's proficiency in reading and reasoning about text embedded in visual content, requiring both visual and textual comprehension to answer questions accurately.

## A.2 Comparison with MLLMs with Multiple Vision Encoders

To better understand how HAWAII compares with the existing MLLMs with multiple vision encoders [13, 14, 47, 68], we present the comparison in Table 6. Results show that HAWAII achieves competitive or significant improvements on most benchmarks, demonstrating the effectiveness of HAWAII. However, we also notice performance degradation on some benchmarks, such as POPE, GQA, and SeedBench.

| | VizWiz | GQA | SQA | POPE | MME | AI2D | MMMU | SeedBench |
|---|---|---|---|---|---|---|---|---|
| Eagle-X5 [47] | **54.4** | **64.9** | 69.8 | **88.8** | 1528 | - | 36.3 | **73.9** |
| MoME [14] (CLIP + DINO + Pix2Struct) | - | 59.7 | - | - | - | - | - | - |
| MouSi [68] (LayoutLMv3+DINOv2+CLIP) | - | 63.6 | 69.0 | 86.5 | - | - | - | 67.5 |
| Brave [13] | 54.2 | 52.7 | - | 87.6 | - | - | - | - |
| LLaVA-1.5 (CLIP) | 50.0 | 62.0 | 66.8 | 85.9 | 1510.7 | 55.5 | 34.7 | 66.1 |
| MoVE-KD | 52.3 | 63.2 | 69.4 | 86.9 | 1524.5 | - | - | - |
| HAWAII | 53.9 | 62.8 | 70.5 | 87.3 | **1540.2** | **56.2** | 36.6 | 67.5 |
| HAWAII† (HAWAII + Pix2Struct) | 54.2 | 63.6 | 69.5 | 86.8 | 1533.6 | **56.2** | 36.0 | 67.2 |
| HAWAII‡ (HAWAII + SAM) | 54.3 | 63.3 | **70.8** | 87.5 | 1528.6 | 55.8 | **36.9** | 67.9 |
| Δ | ↓ 0.1 | ↓ 1.3 | ↑ 1.0 | ↓ 1.3 | ↑ 12.2 | - | ↑ 0.6 | ↓ 6.0 |

Table 6: Comparison with MLLMs with multiple vision encoders.

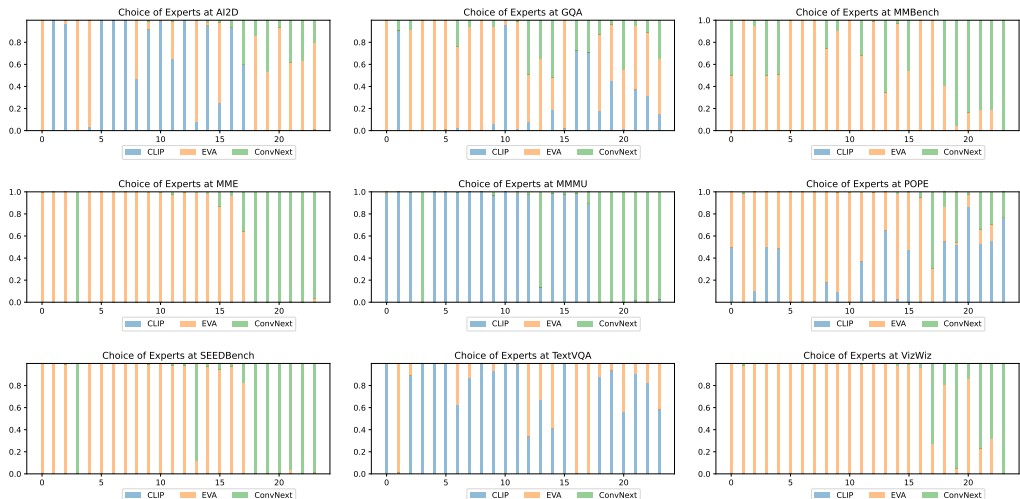

Figure 6: Visualization of the routing choice using HAWAII-v1.0. Best viewed in color.

## A.3 Ablation Study

**Routing between specific teachers' knowledge.** To further understand how HAWAII switches between different teachers' knowledge, we visualize the routing results in Figure 6. It is obvious that HAWAII selects different expert's knowledge across different benchmark datasets and different layers. A notable observation is for most of the cases, HAWAII does not choose CLIP for understanding visual contents. We observe that for MME, VizWiz, and SEEDBench, the model has similar selection preference, while for MMMU, model mainly choose CLIP and ConvNext.

