# OpenReview forum: "Hawaii: Hierarchical Visual Knowledge Transfer for Efficient Vision-Language Models"
_NeurIPS.cc/2025/Conference — NeurIPS 2025 poster_

### Official Review · Reviewer_XygR · 2025-06-01

**Clarity:** 3
**Significance:** 3
**Originality:** 3
**Rating:** 5
**Confidence:** 4

**Summary:**

The article introduced a novel framework that distills knowledge from multiple visual experts into a single vision encoder, enabling it to inherit the complementary strengths of several experts with minimal computational overhead.

**Questions:**

1. I noticed that you have chosen a rank of 16 for your LoRA adapters.  Have you experimented with higher ranks, such as 32, to see if there are any additional performance improvements? Because the MoVE-KD uses the rank of 32.
2. How long does training take?

**Ethical Concerns:**

["NO or VERY MINOR ethics concerns only"]

**Final Justification:**

The authors have provided additional responses and clarifications to all four of my concerns, which effectively address my questions and further validate the proposed method.
As a result, I have increased my score to 5 (accept). The key improvements include:
﻿
* Providing empirical validation with the larger models, which demonstrates the scalability and robustness of the proposed approach in more complex settings.
﻿
* Extending the evaluation to a broader range of models, such as LLaVA-Next, thereby supporting the generalizability of the method.
﻿
* Offering detailed clarification on the choice and impact of hyperparameters.
﻿
* Supplementing information regarding the training time and computational resources involved.

These thoughtful additions significantly strengthen the paper.

**Limitations:**

yes

**Quality:**

3

**Strengths And Weaknesses:**

Strengths：HAWAII proposes a hierarchical knowledge distillation (HKD) mechanism that includes both coarse-grained and fine-grained distillation. This allows for a more nuanced transfer of knowledge from multiple teachers to a single student encoder. In contrast, MoVE-KD primarily focuses on a single level of distillation, which may not capture the full spectrum of knowledge as effectively.
HAWAII introduces the MOLA module, which consists of teacher-specific LoRA adapters and general-knowledge LoRA adapters. This design enables the student encoder to learn from diverse teachers separately (fine-grained) and globally (coarse-grained), avoiding noisy and redundant knowledge transfer.

Weakness：The experiments in the "HAWAII" paper are primarily conducted on the 7B LLaVA-1.5 model. There is a lack of validation on models of different sizes, both larger and smaller than 7B. This limits the generalizability of the findings.  The effectiveness of the HAWAII framework in enhancing visual understanding might vary across different model architectures. Testing on a wider range of models, such as LLaVA-NeXT, would provide a more comprehensive evaluation of the framework's robustness and applicability. (I think this is a very core experiment and cannot be simply summarized in Limitations.)

---

> ### Author Rebuttal · Authors · 2025-07-31
>
> We appreciate your thoughtful comments and valuable feedback.
> Below, please find our specific responses (R) to weaknesses (W) and questions (Q).
>
> ---
>
> > W1: Only test on 7B model.
>
> R: Thank you for the comment. We have added the experimental results on the LLaVA-1.5-13B model as shown below.
> |               | VQA Text | GQA      | SQA    | POPE     | MME        | MMMU | AI2D | SeedBench |
> |---------------|----------|----------|--------|----------|------------|------|------|-----------|
> | LLaVA-1.5-13B | 61.3     | 63.3     | 71.6   | 85.9     | 1531.3     | 35.5 | 59.3 | 68.2      |
> | MoVE-KD-13B   | 59.7     | 64.2     | 73.2   | 85.7     | 1568.1     | -    | -    | -         |
> | **HAWAII-13B (Ours)**    | **61.7** | **64.7** | **75.0** | **86.6** | **1568.7** | **35.7** | **60**   | **68.5**      |
>
> Overall, HAWAII-13B (with CLIP, EVA, and ConvNext as teachers) achieves best performance on all the benchmark datasets, compared to LLaVA-1.5-13B and MoVE-KD-13B [1], which are the two most relevant baselines in our methods. We will add experimental results using smaller models in our final version.
>
> ---
>
> > W2: Testing on a wider range of models.
>
> R: Thank you for the suggestion. Following MoVE-KD [1], which serves as the most relevant and established knowledge distillation baseline for our model, we use LLaVA-1.5 as our base model to ensure a fair comparison. Now, we are employing our methods to LLaVA-Next. The experimental results will be updated in this thread in the following days.
> Moreover, we are also planning to use Qwen-2.5-VL as an additional base model. The results will be added to the final version of our paper.
>
> ---
>
> > Q1: Why rank 16 for LoRA?
>
> R: We use 16 for training efficiency. Due to time limitation, we could not complete the experiments with the LoRA rank of 32. But the corresponding results will be added to the final version for a fairer comparison to MoVE-KD [1].
>
> ---
>
> > Q2: How long does training take?
>
> R: We used 8 A6000 with 48G GPU memory for each. It takes 20 hours for pretraining and 48 hours for supervised fine-tuning on the 7B model. We also tested on 8 A100 with 80G GPU memory for each. It is much faster, with 9 hours for pretraining and 20 hours for fine-tuning. We will add these training details in the final version.
>
> ---
>
> [1] MoVE-KD: Knowledge Distillation for VLMs with Mixture of Visual Encoders. CVPR 2025.

---

> ### Author Response · Authors · 2025-08-02
> **Thank you for your valuable comments. Please let us know if you have any questions.**
>
> Thank you for the thoughtful feedback. We hope our responses have addressed your concerns. **Please don’t hesitate to let us know if you have any remaining questions.**
>
> Also, we collected results using LLaVA-Next as the base model as follows.
>
> ---
> > W2: Testing on a wider range of models (LLaVA-Next-7B).
>
> We have collected the experimental results with LLaVA-Next-7B being the base model, as shown below.
> |                             | VQA Text | GQA      | SQA      | POPE     | MME        | MMMU     | AI2D    |
> |-----------------------------|----------|----------|----------|----------|------------|----------|----------|
> | LLaVA-Next-7B               | 64.9     | 64.2     | 70.1     | 86.5     | 1519.0     | 35.8     | 64.9     |
> | MOVE-KD (LLaVA-Next-7B)     | 63.7     | 64.5     | 70.7     | 86.7     | 1537.2     | -        | -         |
> | HAWAII (Ours-LLaVA-Next-7B) | **65.5** | **65.2** | **72.0** | **87.8** | **1551.3** | **37.4** | **65.6**   |
>
> Our proposed HAWAII shows consistent improvements on most of the benchmark datasets compared to the baseline LLaVA-Next-7B and MOVE-KD (LLaVA-Next-7B), which is the most important baseline method on visual distillation of MLLMs.
> Due to the submission constraints of VizWiz and MMBench, the evaluation cannot be completed at this time. We are also working on implementing HAWAII to Qwen-VL-2.5. We will update the results on those benchmarks and Qwen-VL-2.5 in the final version.

---

> > ### Comment · Reviewer_XygR · 2025-08-04
> >
> > Thank you for providing additional experiments with the larger models, which further validate the effectiveness of your proposed approach. The extra ablation studies and detailed explanations have successfully addressed my concerns.
> > Based on the improvements and clarifications provided, I will increase my score from 4 to 5 (accept). I appreciate the authors’ efforts and contributions to this work.

---

> > > ### Author Response · Authors · 2025-08-05
> > > **Thank you for your time and valuable comments! Please let us know if you have any further questions.**
> > >
> > > Thank you very much for your positive feedback and for taking the time to review our work and response. We're glad that our response has addressed your concerns. We sincerely appreciate your thoughtful comments and your updated score. Please don't hesitate to let us know if you have any further questions or suggestions.

---

### Official Review · Reviewer_p3zt · 2025-06-04

**Clarity:** 2
**Significance:** 2
**Originality:** 2
**Rating:** 4
**Confidence:** 5

**Summary:**

HAWAII is a framework that distills multiple pretrained visual experts into a single vision encoder, capturing their complementary strengths without incurring high computational costs. Instead of relying on a fixed set of adapters for each teacher, HAWAII assigns a dedicated LoRA adapter to each expert and uses a router to switch between them, preventing noisy or conflicting guidance during distillation.

**Questions:**

Refer to Weaknesses

**Ethical Concerns:**

["NO or VERY MINOR ethics concerns only"]

**Final Justification:**

I support this paper's acceptance because the authors did a really great in having a rebuttal.

**Limitations:**

yes

**Quality:**

2

**Strengths And Weaknesses:**

Strenghts:

- Multi-Teacher Knowledge Distillation: HAWAII effectively distills knowledge from multiple pretrained visual experts into a single student encoder, ensuring comprehensive learning from diverse sources.

- Mixture-of-LoRA Adapter (MOLA): The novel MOLA module incorporates both teacher-specific and general-knowledge LoRA adapters, enabling the student encoder to absorb specialized expertise while retaining broad, task-agnostic representations.

- Hierarchical Knowledge Distillation (HKD): HKD operates at two levels—coarse-grained distillation aggregates global knowledge from all teachers, while fine-grained distillation selects the most informative tokens from each teacher—resulting in richer, more targeted guidance.

Weaknesses:

I feel it is great work, but there are some four concerns. First one is that the authors only use LLaVA-1.5 as baseline to apply your method, but this paper could be improved by using multiple recent released models such as Qwen2/2.5-VL and InternVL2.5/3 or the models you are using for evaluation baselines. The second is that there are none of challening benchmarks like MM-Vet, MM-Vet-v2, and MMMU-Pro. Third is that I am wondering why the authors dealt with only LoRA adapter although there are various adapters based on DoRA, VeRA, OFT, LoKr, LoHa, etc. The last thing is that the possibility for LLM-distillation, because the authors only deal with distillation of visual properties but it may be naturally wondering that why didn't the author consider LLM-part distillation.

---

> ### Author Rebuttal · Authors · 2025-07-31
>
> We appreciate your thoughtful comments and valuable feedback.
> Below, please find our specific responses (R) to weaknesses (W) and questions (Q).
>
> ---
>
> > W1: Apply HAWAII to more methods.
>
> R: Thank you for the suggestion. Following MoVE-KD [1], which serves as the most relevant and established knowledge distillation baseline for our model, we use LLaVA-1.5 as our base model to ensure a fair comparison. Now, we are employing our methods to LLaVA-Next. The experimental results will be updated in this thread in the following days.
> Moreover, we are also planning to use Qwen-2.5-VL as an additional base model. The results will be added to the final version of our paper.
>
>
> ---
>
> > W2: Evaluation on challenging benchmarks like MM-Vet, MM-Vet-v2, and MMMU-Pro.
>
> R: Thank you for the suggestion. We have added evaluation on MM-Vet, MMVet-V2, and MMMU-Pro using lmms-eval. It shows that our model HAWAII outperforms LLaVA-1.5-7B on MMMU-Pro, MMVet, and MMVet-2.
> |              | MMMU_pro_standard | MMMU_pro_vision | MMVet | MMVet-2 |
> |--------------|-------------------|-----------------|-------|---------|
> | LLaVA-1.5-7B | 20.3              | 12.1            | 31.1  | 28.3    |
> | HAWAII       | 21.5              | 12.6            | 33.0  | 29.4    |
>
>
> We will test all the models in our paper on these three datasets and the results will be added to the final version. Please let us know if there are any other valuable benchmark datasets. We are more than happy to test our model and report the results.
>
>
> ---
>
> > W3: Why not other LoRA adapters?
>
> R: Thank you for the question. We use LoRA in our design because of its generalizability.
> We have conducted an experiment using DoRA as shown below.
> |                  | VQA Text | GQA      | SQA      | POPE     | MME        | MMMU     | AI2D     | SeedBench |
> |------------------|----------|----------|----------|----------|------------|----------|----------|-----------|
> | LLaVA-1.5-7B     | 58.2     | 62       | 66.8     | 85.9     | 1510.7     | 34.7     | 55.5     | 66.1      |
> | MoVE-KD-7B       | 58.3     | **63.2** | 69.3     | 86.9     | 1524.5     | 35.3     | 55.6     | 66.5      |
> | HAWAII-7B (LoRA) | **58.7** | 62.8     | **70.5** | 87.3 | 1540.2 | **36.6** | **56.2** | 67.5  |
> | HAWAII-7B (DoRA) | 58.4     | 61.8     | 69.3     | **87.7**     | **1558.5**     | 35.2     | 55.2     | **67.8**      |
>
> The results show that, DoRA has comparable performance on most of the benchmark datasets compared with LoRA. It outperforms LoRA on POPE, MME, and AI2D with worse performance on other benchmarks.
> We will add the experiments with more LoRA adapters for testing the scalability of our methods in the final version.
>
>
> ---
>
> > W4: Exploration on LLM distillation.
>
> R: Thank you for the comment. As mentioned in the limitations of our paper, we acknowledge that further improvements might be achieved by LLM distillation.
> In this paper, we focus on visual distillation and propose novel coarse-grained and fine-grained distillation for better transferring knowledge from multiple visual experts into one unified visual encoder.
> We will leave the combination of LLM and visual distillation as future work.
>
> ---
> Reference:
>
> [1] MoVE-KD: Knowledge Distillation for VLMs with Mixture of Visual Encoders. CVPR 2025.

---

> ### Comment · Reviewer_p3zt · 2025-08-01
> **Response**
>
> I've carefully read the rebuttals and thank you for your efforts. The concerns that I was posing initially are mostly addressed. I've updated the scores towards leaning accept. I suggest that authors reflect on not only my comments but also others' ones, and furthermore please make sure your promise is realized through making Qwen2.5-VL's score and other challening benchmarks.

---

> ### Author Response · Authors · 2025-08-02
> **Thank you and this is our experimental results on LLaVA-Next**
>
> Thank you for the thoughtful feedback and updated evaluation. **Please let us know if you have any further questions.**
>
> Also, we collected results using LLaVA-Next as the base model as follows.
>
> ---
> > Additional experimental results on LLaVA-Next-7B.
>
> We have collected the experimental results with LLaVA-Next-7B being the base model, as shown below.
> |                             | VQA Text | GQA      | SQA      | POPE     | MME        | MMMU     | AI2D    |
> |-----------------------------|----------|----------|----------|----------|------------|----------|----------|
> | LLaVA-Next-7B               | 64.9     | 64.2     | 70.1     | 86.5     | 1519.0     | 35.8     | 64.9     |
> | MOVE-KD (LLaVA-Next-7B)     | 63.7     | 64.5     | 70.7     | 86.7     | 1537.2     | -        | -         |
> | HAWAII (Ours-LLaVA-Next-7B) | **65.5** | **65.2** | **72.0** | **87.8** | **1551.3** | **37.4** | **65.6**   |
>
> Our proposed HAWAII shows consistent improvements on most of the benchmark datasets compared to the baseline LLaVA-Next-7B and MOVE-KD (LLaVA-Next-7B), which is the most important baseline method on visual distillation of MLLMs.
> Due to the submission constraints of VizWiz and MMBench, the evaluation cannot be completed at this time. We are also working on implementing HAWAII to Qwen-VL-2.5. We will update the results on those benchmarks and Qwen-VL-2.5 in the final version.

---

> ### Comment · Reviewer_p3zt · 2025-08-07
>
> Looks great to me. Please make sure the final version includes more recently released models

---

### Official Review · Reviewer_A5aZ · 2025-07-03

**Clarity:** 3
**Significance:** 2
**Originality:** 2
**Rating:** 4
**Confidence:** 4

**Summary:**

This paper introduces a new framework to improve the efficiency and performance of VLMs, by distilling knowledge from multiple visual experts into a single vision encoder. The key innovation is the Mixture-of-LoRA-Adapters (MOLA) module, which includes teacher-specific and general-knowledge LoRA adapters. These adapters are selectively activated using sparse routers, allowing the model to learn from diverse teachers without noisy or conflicting guidance. HAWAII performs hierarchical knowledge distillation at both coarse and fine granularity. In fine-grained distillation, the model uses teacher-specific adapters and a novel token importance scoring mechanism to focus on the most informative visual tokens from each teacher. In coarse-grained distillation, it combines features from all teachers into a summarized representation, which is then transferred to the student encoder via general LoRA adapters. Extensive experiments show that HAWAII outperforms previous methods, including MoVE-KD, on various VLM benchmarks like VizWiz, SQA, and MMBench.

**Questions:**

1. How are the teacher models selected?
The paper lacks a clear explanation of the rationale behind selecting CLIP, ConvNeXt, and EVA-02 as the teacher models. It is unclear whether these models were chosen due to their complementary strengths, diverse training objectives, or empirical performance.
2. Is knowledge distillation applied during both pretraining and fine-tuning stages?
While the training pipeline is described, the paper does not clearly state whether the hierarchical knowledge distillation is performed in both the pretraining and SFT stages, or only in one.

**Ethical Concerns:**

["NO or VERY MINOR ethics concerns only"]

**Final Justification:**

The author's rebuttal has addressed my concerns, and I will raise the rating.

**Limitations:**

yes

**Quality:**

2

**Strengths And Weaknesses:**

Strengths:
1. The paper is well-written, logically structured, and easy to follow. The authors provide a detailed explanation of their method and offer comprehensive illustrations to aid understanding.
2. The paper addresses a meaningful and practical problem in the vision-language domain—how to efficiently integrate knowledge from multiple visual experts into a single model. This is relevant to the community, especially for resource-constrained settings.

Weaknesses:
1. Limited Novelty. While the paper proposes using teacher-specific LoRA adapters and hierarchical knowledge distillation, these techniques are already widely used in other domains.
2. Insufficient Experiments. The experimental validation lacks critical comparisons. In Table 1, there is no baseline that uses multiple visual encoders at inference time (e.g., model ensembles or token concatenation), which makes it difficult to quantify the effectiveness of the distillation process. Moreover, the ablation studies are limited. The authors should include comparisons showing how different vision experts (e.g., CLIP, SAM, Pix2Struct) impact the final student model’s performance when used individually or in combination.
3. Outdated Baselines. Many of the compared models (e.g., BLIP-2, LLaVA-1.5) are early-stage VLMs. The paper does not compare against more recent and stronger models like Qwen-VL 2.5 and InternVL 2.5. Given that the method aims to enhance vision encoders via multi-teacher distillation, it would be important to test whether the proposed method can still improve state-of-the-art VLMs when more advanced vision backbones are used.

---

> ### Author Rebuttal · Authors · 2025-07-31
>
> We appreciate your thoughtful comments and valuable feedback.
> Below, please find our specific responses (R) to weaknesses (W) and questions (Q).
>
> ---
>
> > W1: Novelty.
>
> R: Thank you for the comments.
> In this paper, we propose HAWAII that distills knowledge from multiple visual experts into a single vision encoder.
> To mitigate conflicts among different teachers and switch between different teacher-specific knowledge, we introduce teacher-specific LoRA adapters, each corresponding to one specific visual teacher.
> We also include token importance scores to emphasize the most informative tokens from each teacher adaptively.
> Moreover, we also employ a set of general-knowledge LoRA adapters to learn the common knowledge among different teachers and training data.
>
> We would like to clarify that although the term hierarchical knowledge distillation has been used in other domains (e.g., [3, 4]), the underlying techniques differ significantly.
> Their methods focus on aligning features extracted from the hierarchical architecture of one teacher, such as features from the shallow layers to the deep layers.
> On the other side, our proposed method, HAWAII, treats hierarchical knowledge distillation as fine-grained (distilling knowledge from one teacher only for solving conflicts between teachers) and coarse-grained knowledge distillation (distilling knowledge from the summarized knowledge of multiple teachers), which is totally different from those methods in other domains.
>
> To the best of our knowledge, we are the first to use two sets of LoRA adapters to distill knowledge from multiple visual experts into one visual encoder. We might miss some references in relevant areas. Please let us know if there are any works that have a similar design of two sets of adapters for distilling knowledge.
>
> ---
>
> > W2: No baseline on models with multiple visual encoders.
>
> R:
> While we appreciate the importance of such baselines with multiple experts [1], our method fundamentally differs in design: we distill knowledge to a single unified vision encoder, rather than maintaining multiple experts during both training and inference.
> Directly comparing these two paradigms would therefore not be fair, as the former with multiple experts typically involves higher computational cost and a larger number of parameters. We have collected the latency of LLaVA-1.5-7B, Eagle-X4-Plus (with four visual experts), and our HAWAII-7B as shown below. We use Pytorch Profiler to measure the FLOPs and CUDA time using the teaser figure from the llava paper and “What is shown in this image?” as input. We use one A6000 GPU with 48 GB GPU memory to inference the models.
> |                  | FLOPs (T) | CUDA Time (s) |
> |------------------|-----------|---------------|
> | LLaVA-1.5-7B     | 8.9       | 1.2           |
> | Eagle-X4-Plus    | 29.4      | 2.1           |
> | HAWAII-7B (ours) | 9.3       | 1.3           |
>
> It shows that, our HAWAII, which only uses one vision encoder, has smaller FLOPs and CUDA times compared to the Eagle model, which contains multiple (4-5) visual experts.
> We will explicitly highlight this distinction in the paper and include a discussion in the final version.
>
> ---
>
> > W2: No ablations on how different vision experts affect performance.
>
> R: Thank you for the comment. We have presented an ablation study on different settings of visual teachers in Table 1 of our paper. It shows that having more visual teachers improves performance, and due to the heterogeneous knowledge in different visual teachers, the performance boost in different benchmark datasets varies.
> For performance using different visual experts [1], we have included the following table for your reference.
> |                               | VizWiz   | GQA      | SQA      | POPE     | MME        | AI2D     | SeedBench |
> |-------------------------------|----------|----------|----------|----------|------------|----------|-----------|
> | LLaVA-1.5 (CLIP as the visual encoder)              | 50       | 62       | 66.8     | 85.9     | 1510.7     | 55.5     | 66.1      |
> | LLaVA-1.5* (SAM as the visual encoder)              | 49       | 57.3     | 66.8     | 84.3     | 1216       | **69.2** | 56.9      |
> | HAWAII (Ours, CLIP as the visual encoder)                 | 53.9     | 62.8     | 70.5     | 87.3     | **1540.2** | 56.2     | 67.5      |
> | HAWAII† (HAWAII + Pix2Struct as teachers) | 54.2     | 63.6     | 69.5     | 86.8     | 1533.6     | 56.2     | 67.2      |
> | HAWAII‡ (HAWAII + SAM as teachers)        | **54.3** | **63.3** | **70.8** | **87.5** | 1528.6     | 55.8     | **67.9**  |
>
> Our HAWAII uses CLIP as the student encoder with CLIP, ConvNext, and EVA being the teachers. Our HAWAII achieves best performance on all the benchmark datasets except AI2D. *represent results from [1].
> We will include a more detailed analysis in the final version.
>
> ---
> > W3: Apply HAWAII to more methods.
>
> R: Thank you for the suggestion. Following MoVE-KD [2], which serves as the most relevant and established knowledge distillation baseline for our model, we use LLaVA-1.5 as our base model to ensure a fair comparison. Now, we are employing our methods to LLaVA-Next. The experimental results will be updated in this thread in the following days.
> Moreover, we are also planning to use Qwen-2.5-VL as an additional base model. The results will be added to the final version of our paper.
>
> ---
>
> > Q1: How are the teacher models selected?
>
> R: Thank you for the question. We select CLIP, ConvNeXt, and EVA-02 as our main design, following MoVE-KD [2], which serves as the most relevant and established knowledge distillation baseline for our model to ensure a fair comparison. We also include Pix2Struct and SAM for analyzing the impact of different visual experts, as shown in Table 1.
>
> Moreover, they serve different purposes: CLIP (general-purpose), ConvNeXt (image classification), Pix2Struct (OCR/document understanding), EVA-02 (multi-task generalist), and SAM (segmentation). These teachers differ in architecture, training data, and task specialization, providing a broad and complementary set of capabilities. And this diversity is intended to balance generality with scalability. Such a diverse set of visual teachers brings comprehensive knowledge from different domains, benefiting the model. We will clarify this in our paper.
>
> ---
>
> > Q2: Is knowledge distillation applied during both pretraining and fine-tuning stages?
>
> R: Yes. It is applied in both pretraining and supervised fine-tuning stages.
> We will clarify this in the paper.
>
> ---
>
> Reference:
>
> [1] Eagle: Exploring The Design Space for Multimodal LLMs with Mixture of Encoders. ICLR 2025.
>
> [2] MoVE-KD: Knowledge Distillation for VLMs with Mixture of Visual Encoders. CVPR 2025.
>
> [3] SignKD: Multi-modal Hierarchical Knowledge Distillation for Continuous Sign Language Recognition.
>
> [4] Hierarchical Self-supervised Augmented Knowledge Distillation. arXiv:2107.13715.

---

> ### Author Response · Authors · 2025-08-02
> **Thank you for your valuable comments. Please let us know if you have any further questions.**
>
> Thank you for the thoughtful feedback. We hope our responses have addressed your concerns. **Please don’t hesitate to let us know if you have any remaining or further questions.**
>
> Also, we collected results using LLaVA-Next as the base model as follows.
>
> ---
> > W3: Apply HAWAII to more methods (LLaVA-Next-7B).
>
> We have collected the experimental results with LLaVA-Next-7B being the base model, as shown below.
> |                             | VQA Text | GQA      | SQA      | POPE     | MME        | MMMU     | AI2D    |
> |-----------------------------|----------|----------|----------|----------|------------|----------|----------|
> | LLaVA-Next-7B               | 64.9     | 64.2     | 70.1     | 86.5     | 1519.0     | 35.8     | 64.9     |
> | MOVE-KD (LLaVA-Next-7B)     | 63.7     | 64.5     | 70.7     | 86.7     | 1537.2     | -        | -         |
> | HAWAII (Ours-LLaVA-Next-7B) | **65.5** | **65.2** | **72.0** | **87.8** | **1551.3** | **37.4** | **65.6**   |
>
> Our proposed HAWAII shows consistent improvements on most of the benchmark datasets compared to the baseline LLaVA-Next-7B and MOVE-KD (LLaVA-Next-7B), which is the most important baseline method on visual distillation of MLLMs.
> Due to the submission constraints of VizWiz and MMBench, the evaluation cannot be completed at this time. We are also working on implementing HAWAII to Qwen-VL-2.5. We will update the results on those benchmarks and Qwen-VL-2.5 in the final version.

---

> > ### Comment · Reviewer_A5aZ · 2025-08-06
> >
> > The authors' rebuttal has addressed most of my concerns.
> >
> > One remaining point pertains to the absence of baselines using multiple visual encoders. I acknowledge that such models would incur higher computational costs. However, their inclusion is crucial as a baseline to effectively measure the efficacy of knowledge distillation for LVLM. I do not require HAWAII to outperform these multi-encoder baselines. Rather, the observed performance gap would indicate the potential for further improvement of this technique.
> >
> > If the authors address this concern, I will raise the rating.

---

> > ### Author Response · Authors · 2025-08-06
> > **Follow-Up for Reviewer A5aZ: Comparison with baselines with multiple visual encoders**
> >
> > Thank you for your positive feedback. We are delighted that our previous response addressed most of your concerns.
> >
> > ---
> >
> > > Regarding the remaining point about baselines with multiple visual encoders:
> >
> > We appreciate your suggestion and fully agree that such comparisons are valuable for understanding the potential and limitations of knowledge distillation for LVLMs.
> >
> > In response, we have included results for several popular LVLMs with multiple vision encoders, i.e., Eagle [1], MoME (CLIP + DINO + Pix2Struct) [2], MouSi (LayoutLMv3 + DINOv2 + CLIP) [3], and Brave [4], in the following tables:
> >
> > *LVLM with multiple vision encoders*
> >
> > |                                     | SFT data   | FLOPs (T) | CUDA Time (s) | VizWiz   | GQA      | SQA      | POPE     | MME        | AI2D | MMMU     | SeedBench |
> > |-------------------------------------|------------|-----------|---------------|----------|----------|----------|----------|------------|------|----------|-----------|
> > | Eagle-X5 [1]                        | Eagle-1.8M | 29.6      | 2.2           | **54.4**     | **64.9** | **69.8** | **88.8** | **1528**   | -    | **36.3** | **73.9**  |
> > | MoME (CLIP + DINO + Pix2Struct) [2] | -          | -         | -             | -        | 59.7     | -        | -        | -          | -    | -        | -         |
> > | MouSi (LayoutLMv3+DINOv2+CLIP) [3]  | -          | -         | -             | -        | 63.6     | 69.0     | 86.5     | -          | -    | -        | 67.5      |
> > | Brave [4]                           | -          | -         | -             | 54.2     | 52.7     | -        | 87.6     | -          | -    | -        | -         |
> >
> > *LLaVA or KD-based LVLM*  (Differences represents the performance between the best version of HAWAII and the best LVLM with multiple vision encoders)
> >
> > |                               | SFT data   | FLOPs (T) | CUDA Time (s) | VizWiz   | GQA      | SQA      | POPE     | MME        | AI2D     | MMMU     | SeedBench |
> > |-------------------------------|------------|-----------|---------------|----------|----------|----------|----------|------------|----------|----------|-----------|
> > | LLaVA-1.5 (CLIP)              | LLaVA data | 8.9       | 1.2           | 50.0       | 62       | 66.8     | 85.9     | 1510.7     | 55.5     | 34.7     | 66.1      |
> > | MoVE-KD                 | LLaVA data | -         | -             | 52.3     | 63.2     | 69.4     | 86.9     | 1524.5     | -        | -        | -         |
> > | HAWAII (Ours)                 | LLaVA data | 9.3       | 1.3           | 53.9     | 62.8     | 70.5     | 87.3     | **1540.2** | **56.2** | 36.6     | 67.5      |
> > | HAWAII† (HAWAII + Pix2Struct) | LLaVA data | 9.4       | 1.3           | 54.2     | **63.6** | 69.5     | 86.8     | 1533.6     | **56.2** | 36.0     | 67.2      |
> > | HAWAII‡ (HAWAII + SAM)        | LLaVA data | 9.4       | 1.3           | **54.3** | 63.3     | **70.8** | **87.5** | 1528.6     | 55.8     | **36.9** | **67.9**  |
> > | Differences                   |            |           |               | -0.1     | -1.3     | +1.0     | -1.3     | +12.2      | -        | +0.6     | -6.0      |
> >
> > Notably, while EAGLE achieves the strongest results on VizWiz, SQA, POPE, and SeedBench, our proposed HAWAII model achieves competitive or even superior performance on SQA, MME, and MMMU, with significantly lower computational cost (FLOPs and CUDA time) compared to multi-encoder models.
> > This demonstrates the effectiveness of our knowledge distillation approach and also highlights the remaining performance gap, which points toward promising directions for future work in the design of LVLM.
> >
> > We will make this comparison and its implications clearer in our final version. If there is any other relevant models or recent works, please let us know and we would be happy to include them.
> >
> > ---
> >
> > Thank you again for your thoughtful feedback and time.
> > *We hope our response has addressed all your concerns. If there are any remaining questions, we are happy to engage in further discussion and clarify them.*
> >
> > Reference:
> >
> > [1] Eagle: Exploring The Design Space for Multimodal LLMs with Mixture of Encoders. ICLR 2025.
> >
> > [2] MoME: Mixture of Multimodal Experts for Generalist Multimodal Large Language Models. NeurIPS 2024.
> >
> > [3] Poly-Visual-Expert Vision-Language Models. COLM 2024.
> >
> > [4] BRAVE: Broadening the Visual Encoding of Vision-Language Models. ECCV 2024.

---

### Official Review · Reviewer_7ZEx · 2025-07-04

**Clarity:** 2
**Significance:** 2
**Originality:** 2
**Rating:** 4
**Confidence:** 3

**Summary:**

This  paper introduces HAWAII, a hierarchical visual knowledge distillation framework that transfer knowledge from multiple pre-trained vision foundation models such as ConvNeXt, EVA, Pix2Struct and SAM, into a single vision encoder to improve efficiency. HAWAII utilizes teacher-specific LoRA adapters and a mixture-of-LoRA-adapters module to align multiple teachers and student. In the while, a hierarchical knowledge distillation mechanism is proposed to combine global and local information to optimize knowledge transfer. Extensive experiments have demonstrated its superior performance across vision-language tasks compared to baselines.

**Questions:**

1. In Table 1, when adding teachers like Pix2Struct, performance on some tasks (e.g., VQA$^{Text}$) slightly decreases due to knowledge redundancy, but the paper does not quantify the optimal number of teachers to avoid such trade-offs.

2. What is the optimal number of LoRA adapters for balancing performance and computation?

3. How does HAWAII perform with domain-specific visual experts (e.g., medical or autonomous driving models)? Further, can the hierarchical distillation mechanism be applied to multi-modal knowledge beyond vision?

**Ethical Concerns:**

["NO or VERY MINOR ethics concerns only"]

**Limitations:**

yes

**Quality:**

2

**Strengths And Weaknesses:**

## Strengths
- A good ides is proposed in this paper, i.e., hierarchical distillation: fine-grained (token-level) and coarse-grained (global consensus) strategies ensure comprehensive knowledge transfer.
- Significantly improves on benchmarks like VizWiz (+7.8) and SQA (+5.5) compared to LLaVA-1.5 baseline.


## Weakesnnes
- While teacher-specific LoRA adapters aim to avoid conflicts, the paper does not explicitly analyze how adapters for different teachers interact during training, leaving uncertainty about potential interference effects.

- The performance of HAWAII on transfer tasks (e.g., adapting from synthetic data to natural images) is unaddressed, limiting understanding of its generalization beyond training distributions.

- The proposed HAWAII focuses solely on visual expert distillation and does not explore integrating knowledge from non-visual modalities (e.g., audio or text).

---

> ### Author Rebuttal · Authors · 2025-07-31
>
> We appreciate your thoughtful comments and valuable feedback.
> Below, please find our specific responses (R) to weaknesses (W) and questions (Q).
>
> ---
>
> > W1: How do adapters for different teachers interact during training.
>
> R: Thank you for the question. During training, we activate one LoRA expert at a time to get the specific student feature $I^{S}_{i}, \forall i \in [N_t]$ for fine-grained distillation (Sec. 2.3.2).
> Also, for calculating coarse-grained distillation loss and text generation loss, only one teacher-specific LoRA adapter and one general-knowledge adapter are activated based on the routers (Eq. (2)).
> Overall, each teacher-specific LoRA adapter will be updated to align with its corresponding teacher independently in fine-grained distillation (Sec. 2.3.2), while coarse-grained distillation (Sec. 2.3.1) will update one teacher-specific LoRA adapter selected by the router.
> That means they will not interfere with each other during training.
>
> ---
>
> > W2: Generalization beyond training distributions.
>
> R: Thank you for the suggestion. Following previous works [1,2], in our experiments, we use LLaVA-1.5 pretraining and SFT data for training the model, which means we report zero-shot performance on most of the benchmark datasets.
> Meanwhile, we are also interested in having more evaluations on synthetic data. Please let us know if there are any relevant, valuable benchmarks we could test our models on.
>
> ---
>
> > W3 and Q3: No exploration on non-visual distillation.
>
> R: Thank you for the comment. As mentioned in the limitations of our paper, we acknowledge that further improvements might be achieved by LLM distillation.
> In this paper, we focus on visual distillation and propose novel coarse-grained and fine-grained distillation to better transfer knowledge from multiple visual experts into one unified visual encoder.
> We will leave the combination of LLM and visual distillation as future work.
>
> ---
>
> > Q1: Optimal number of visual teachers.
>
> R: In Table 1 of our paper and the following table, we did observe that under different settings of the teachers, the model performs differently on diverse benchmark datasets, which might be due to the diverse knowledge brought by teachers.
> Thus, instead of finding an optimal number or set of visual teachers for all the diverse tasks, we suggest that users select the teachers that best suit the downstream tasks for better performance.
>
> |                               | VizWiz   | GQA      | SQA      | POPE     | MME        | AI2D     | SeedBench |
> |-------------------------------|----------|----------|----------|----------|------------|----------|-----------|
> | LLaVA-1.5 (CLIP as the visual encoder)              | 50       | 62       | 66.8     | 85.9     | 1510.7     | 55.5     | 66.1      |
> | HAWAII (Ours, CLIP as the visual encoder)                 | 53.9     | 62.8     | 70.5     | 87.3     | **1540.2** | **56.2**     | 67.5      |
> | HAWAII† (HAWAII + Pix2Struct as teachers) | 54.2     | 63.6     | 69.5     | 86.8     | 1533.6     | **56.2**     | 67.2      |
> | HAWAII‡ (HAWAII + SAM as teachers)        | **54.3** | **63.3** | **70.8** | **87.5** | 1528.6     | 55.8     | **67.9**  |
>
> ---
>
> > Q2: Optimal number of LoRA adapters.
>
> R: The number of teacher-specific LoRA adapters is dependent on the number of visual teachers.
> And the number of general-knowledge LoRA adapters is determined by the users.
> We have added an ablation on the number of general-knowledge LoRA adapters, as shown below
>
> | # of General-Knowledge adapters | VQA Text | GQA      | SQA      | POPE     | MME        | MMMU     | AI2D     | SeedBench |
> |--------------------------------|----------|----------|----------|----------|------------|----------|----------|-----------|
> | 1                              | **58.7** | 62.6     | 70.1     | 84.5     | 1516.2     | **37.0** | 55.5     | 67.4      |
> | 3 (Main Configuration)         | **58.7** | **62.8** | **70.5** | **87.3** | **1540.2** | 36.6     | **56.2** | **67.5**  |
> | 5                              | 58.6     | **62.8** | 70.4     | 85.2     | 1530.2     | 36.4     | 55.0       | 66.9      |
>
> In our paper, we use 3 general-knowledge experts as the main setting for HAWAII.
> Results show that, the setting of 3 general-knowledge LoRA adapters outperforms other settings with 1 and 5 adapters on most of the benchmark datasets.
> We also observe that on MMMU, increasing the number of general-knowledge LoRA adapters decreases performance, while on GQA, using 3 or 5 adapters works the best.
>
> ---
>
> > Q3: Include domain-specific visual experts.
>
> R: Thank you for the suggestions. In our paper, following MoVE-KD [2] and Eagle [3], we use CLIP, EVA, and ConvNext as the basic settings of teachers, with Pix2Struct and SAM as additional teachers.
> They serve different purposes: CLIP (general-purpose), ConvNeXt (image classification), Pix2Struct (OCR/document understanding), EVA-02 (multi-task generalist), and SAM (segmentation).
> These teachers differ in architecture, training data, and task specialization, providing a broad and complementary set of capabilities. And this diversity is intended to balance generality with scalability. Such a diverse set of visual teachers brings comprehensive knowledge from different domains, benefiting the model.
> Moreover, we would like to explore some other choices for the visual experts, such as medical or driving models.
> Please let us know if there are any specific models you would suggest.
>
> ---
>
> Reference:
>
> [1] Visual Instruction Tuning. NeurIPS 2023.
>
> [2] MoVE-KD: Knowledge Distillation for VLMs with Mixture of Visual Encoders. CVPR 2025.
>
> [3] Eagle: Exploring The Design Space for Multimodal LLMs with Mixture of Encoders. ICLR 2025.

---

> > ### Author Response · Authors · 2025-08-09
> > **A Follow-up**
> >
> > Dear Reviewer 7ZEx,
> >
> > Thank you again for your valuable feedback and comments. We hope our previous response addressed all your concerns. As the discussion period is nearing its end, we would greatly appreciate it if you could share any remaining questions or thoughts. We would be happy to clarify or provide additional information.
> >
> > Thank you again for your time and effort.
> >
> > Best regards,
> > Authors

---

### Author Response · Authors · 2025-08-09
**General Follow-up**

We sincerely thank all reviewers for their time and valuable feedback.

During the rebuttal and discussion periods, we made significant efforts to provide detailed responses, including additional experiments and clarifications, to address the reviewers’ concerns comprehensively.

As of now, three out of four reviewers have confirmed that their concerns were addressed, and we look forward to hearing from the remaining reviewer should they have any additional questions. Moreover, we are encouraged that multiple reviewers have recognized the strengths of our work, including the novelty and efficiency of the proposed HAWAII, the experimental results, the quality of writing, and the potential impact.

Best regards,

Authors

---

### Note · Authors · 2025-08-16

Dear Area Chair and Reviewers,

We sincerely thank you for the thorough and constructive feedback. We are encouraged that discussions with Reviewers A5aZ, p3zt, and XygR successfully addressed their concerns, and no new questions have arisen from the discussion stage.

For your final consideration, we wish to summarize our responses to the key points:

- We have clarified comments regarding training times, the selection of vision teachers, the comparison to knowledge distillation methods, the comparison to MLLMs with multiple vision encoders, and the interaction between different teachers during training.
- We have added additional experiments exploring the impact of the number of visual teachers, the number of LoRA adapters, other LoRA methods, results on challenging benchmarks, using a large base model (LLaVA-1.5-13B), and applying our HAWAII to LLaVA-Next.

Moreover, we are encouraged that multiple reviewers have recognized the strengths of our work, including the novelty and efficiency of the proposed HAWAII, the experimental results, the quality of writing, and the potential impact.

*We are fully prepared to present the strongest version of our work, shaped by the reviewers' valuable feedback.* We sincerely thank you for your leadership and coordination, as well as all the reviewers for their time and insightful comments.

Warmest regards,

Authors of Submission 18182

---

### Decision · Program_Chairs · 2025-09-17

**Decision:**

Accept (poster)

**Comment:**

The paper presents a practical and well-executed approach to multi-expert visual distillation, featuring clear architectural choices and hierarchical objectives that yield consistent gains across benchmarks and model sizes, while maintaining efficiency relative to multi-encoder systems. Novelty is moderate, and some evaluations are pending (additional base models and LoRA rank sweep); however, the rebuttal added substantial evidence, addressed key methodological questions, and provided efficiency comparisons.